# Synthesis and Bioactivity of Hydrazide-Hydrazones with the 1-Adamantyl-Carbonyl Moiety

**DOI:** 10.3390/molecules24214000

**Published:** 2019-11-05

**Authors:** Van Hien Pham, Thi Phuong Dung Phan, Dinh Chau Phan, Binh Duong Vu

**Affiliations:** 1Drug R&D Center, Vietnam Military Medical University. No.160, Phung Hung str., Phuc La ward, Ha Dong district, Hanoi 100000, Vietnam; phamvanhien181288@gmail.com; 2Department of Pharmaceutical Chemistry, Hanoi University of Pharmacy. No. 15, Le Thanh Tong Str., Hoan Kiem district, Hanoi 100000, Vietnam; pdungdhd@gmail.com; 3Hanoi University of Science and Technology. No.1, Dai Co Viet str., Bach Khoa ward, Hai Ba Trung district, Hanoi 100000, Vietnam

**Keywords:** adamantane derivatives, hydrazide-hydrazone, antimicrobial, cytotoxicity activity

## Abstract

Reaction of 1-adamantyl carbohydrazide (**1**) with various substituted benzaldehydes and acetophenones yielded the corresponding hydrazide-hydrazones with a 1-adamantane carbonyl moiety. The new synthesized compounds were tested for activities against some Gram-negative and Gram-positive bacteria, and the fungus *Candida albicans*. Compounds **4a**, **4b**, **5a**, and **5c** displayed potential antibacterial activity against tested Gram-positive bacteria and *C. albicans*, while compounds **4e** and **5e** possessed cytotoxicity against tested human cancer cell lines.

## 1. Introduction

The hydrazide-hydrazones derivatives, which play an important role in organic and medicine chemistry, have attracted a large number of researchers over the years because of their promising biological activities, including antimicrobial [1,2,3,4], anticancer [5,6,7], antituberculosis [2,8,9,10], antiviral [11], and anticonvulsant [2] activities. Some hydrazide-hydrazone components have been considered as drugs and have been used in clinics, such as nitrofurazone, furazolidone, nitrofurantoin.

Adamantane derivatives are important constituents in medicinal chemistry due to their multiple bioactivities. In the 1960s, the first adamantane derivative, amantadine, was found to have antiviral [12,13,14] activity. A number of studies were conducted, focusing on such derivatives with the hope to find new compounds and biological activities. As a result, many adamantane derivatives were discovered with numerous biological activities, such as antiviral [15,16,17,18,19,20], antimicrobial [18,20,21,22,23,24,25,26,27,28,29,30], anticancer [23,27,31,32,33,34,35], angiogenesis inhibition [36], anti-inflammation [37], 11β hydroxysteroid dehydrogenase type 1 (11β-HSD1) inhibition [38,39,40], and tyrosinase inhibition [41] activities. Among them, nine compounds were approved for use in clinics, including amantadine, memantine, rimantadine, tromantadine, adapalene, vildagliptine, saxagliptine, bromanatane, and adapromine. Therefore, the combination of two moieties, hydrazone and adamantane, had potential to confer new molecules with promising biological activities. In this study, we report the synthesis and biological activities of hydrazide-hydrazones with the 1-adamantyl carbonyl moiety.

## 2. Results and Discussion

### 2.1. Chemistry

In this study, adamantane-1-carbohydrazide was used as the key intermediate. It was initially prepared by esterification of 1-adamantane carboxylic acid (**1**) and methanol under catalysis of 98% H_2_SO_4_ to yield methyl ester **2**. Then, ester **2** was reacted with hydrazine hydrate to yield adamantyl-1-carbohydrazide (**3**). Subsequently, compound **3** was condensed with aromatic aldehydes and ketones to yield the corresponding hydrazide-hydrazones **4a**–**i** and **5a**–**k**, as seen in Scheme 1 and Table 1. The structure of hydrazide hydrazones **4a**–**i** and **5a**–**k** was confirmed by ^1^H-NMR, ^13^C-NMR, and electron impact (ESI-MS) mass spectral data.

### 2.2. In Vitro Antimicrobial Activity

The newly synthesized hydrazide-hydrazones **4a**–**i** and **5a**–**k** were tested for their in vitro growth inhibition against standard strains of the National Institute for Food Control (NIFC, Vietnam), including three Gram-negative bacteria, including *Escherichia coli* (ATCC25922), *Pseudomonas aeruginosa* (ATCC27853), and *Salmonella enterica* (ATCC12228); three Gram-positive bacteria, including *Enterococcus faecalis* (ATCC13124), *Staphylococcus aureus* (ATCC25923), and *Bacillus cereus* (ATCC 13245); and one yeast-like pathogenic fungus, *Candida albicans* (ATCC10231). The primary screening was conducted using the microplate dilution method, which utilized Luria-Bertani (LB) broth medium. The antibiotic Streptomycin and antifungal drug Cyloheximide were used as the positive samples. The results of preliminary antimicrobial testing of the newly synthesized hydrazide-hydrazones **4a**–**i** and **5a**–**k** are presented in Table 2 and Table 3. 

The results of antimicrobial testing showed that all newly synthesized hydrazide-hydrazones possessed various inhibition effects on the tested Gram-positive bacteria and yeast-like fungus from moderate to serious activities. Strong antibacterial activity was found for compounds **4a**, **4d**, **5a**, and **5c**. Meanwhile, compound **4a** strongly inhibited EF, SA, BC, and CA, with an MIC of 12.5 μM (for the four strains) and IC_50_ values of 6.35 μM, 6.77 μM, 6.12 μM, and 6.37 μM, respectively. Additionally, compounds **4d**, **5a**, and **5c** have good inhibitory effects on EF and CA, with MICs of 12.5 μM and 6.26 μM, 12.5 μM and 12.5 μM, 12.5 μM and 12.5 μM; and IC_50_ values of 6.88 μM and 3.56 μM, 6.73 μM and 6.25 μM, and 6.77 μM and 6.66 μM, respectively. Moreover, compounds **4f** and **4h** showed good results in inhibition of the yeast-like fungus CA with MICs of 12.5 μM (for both synthesized compounds) and IC_50_ values of 6.77 μM and 6.45 μM, respectively. Other synthesized compounds showed moderate inhibition of the tested Gram-positive bacteria and *C. albicans*.

Regarding the structure–antimicrobial activity relationship of the synthesized compounds, inhibition increased significantly for compounds possessing an –OH (*para*) group on the benzyl ring (compounds **4a** and **5a**). Within the hydrazide-hydrazones **4a**–**i**, it was suggested that the inhibition against EF did not change substantially when having –NO_2_, –OC_2_H_5_, or an –OCH_3_ group at position 4 of the benzyl ring, but decreased if they were replaced by a –Br group. Otherwise, the presence of only one substituent on the benzyl ring inhibited SA better than that of two substituents. In the inhibition against BC, it seemed that the presence of 4-OH, 4-NO_2_, 4-OC_2_H_5_ enhanced the activity; however, it was reduced when the substituent was a 4-OCH_3_ group. In addition, the inhibition of CA showed a good result while having only one substituent of 4-NO_2_, 4-OC_2_H_5_, or 4-OCH_3_ on the benzyl ring.

In hydrazide-hydrazones **5a**–**k**, it seemed that the presence of the benzyl ring enhanced the antimicrobial activity. Compound **5k**, containing two methyl groups (2-CH_3_ and 5-CH_3_), showed good inhibition against Gram-positive bacteria and the yeast-like fungus *C. albicans*. The replacement of 2-CH_3_ by 2-OH enhanced the activity, particularly against *C. albicans*, but made no improvement to the inhibition of Gram-positive bacteria.

### 2.3. In Vitro Cytotoxicity

All the synthesized hydrazide-hydrazones **4a**–**i** and **5a**–**k** were evaluated for their cytotoxicity in four human cancer cell lines, including hepatic cancer cell line Hep3B, human cervical cancer cell line HeLa, human lung cancer cell line A549, and human breast carcinoma MCF-7, in line with a previous publication [42]. As shown in Table 4, some of the synthesized hydrazide-hydrazones exhibited antiproliferative activity against the tested cancer cell lines. Among them, compound **5e** demonstrated the most potent cytotoxicity on cell viability of 37.78 ± 2.44%, 40.42 ± 0.38%, 19.62 ± 1.74%, and 34.13 ± 2.22% in HeP3B, Hela, A549, and MCF-7 cell lines, respectively, at a concentration of 100 μM. In addition, compound **4e** inhibited Hela, A549, and MCF-7 cells, with cell viabilities of 44.37 ± 1.39%, 38.51 ± 1.59%, and 38.69 ± 1.20% at a concentration of 100 μM, respectively. Other compounds showed slight cytotoxicity on the tested cancer cell lines. Structure–cytotoxicity relationship analysis revealed that compounds **4e** and **5e**, containing both 3-NO_2_ and 4-Cl groups on the benzyl ring, have significant cytotoxicity against the four tested cancer cell lines.

## 3. Experimental

### 3.1. General

Melting point (°C) was measured in a glass microtube using a MP50 Mettler Teledo electrothermal melting point apparatus. NMR spectra were obtained with a Bruker Biospin AVANCE III spectrometer system monitoring at 500 MHz in DMSO-*d*_6_; the chemical shifts (δ, ppm) were expressed and coupling constants (J) were given in Hz using tetramethylsilane as the internal standard. Mass spectra (MS) were recorded on an Agilent 910-TQ-FT-MS system. Monitoring the reactions and preliminary purity checking of synthesized compounds were conducted by thin layer chromatography (TLC) using precoated aluminum sheet 60 F254 plates (Merck KGaA Co., Darmstadt, Germany; EMD Milipore Corp., MA, USA) and visualization with UV at 254 nm. The bacterial and fungus strains were supplied by National Institute for Food Control (NIFC, Hanoi, Vietnam) including *Escherichia coli* (ATCC25922), *Pseudomonas aeruginosa* (ATCC27853), *Salmonella enterica* (ATCC12228), *Enterococcus faecalis* (ATCC13124), *Stapphylococus aureus* (ATCC25923), *Bacillus cereus* (ATCC 13245), and *Candida albicans* (ATCC10231). This study was conducted on cancer cell lines supplied by Advanced Center for Bioorganic Chemistry (ACBC) of the Institute of Marine Biochemistry (IMBC), Vietnam Academy Science and Technology (VAST), i.e., hepatic cancer cell line Hep3B, human cervical cancer cell line HeLa, human lung cancer cell line A549, and human breast carcinoma MCF-7.

### 3.2. Synthesis of Methyl Adamantane-1-Carboxylate (***2***)

A mixture of 5 g (2.7 mmoL) adamantane-1-carboxylic acid, 50 mL methanol (1235 mmoL), and 9.2 g H_2_SO_4_ 98% was stirred under reflux conditions. The reaction was monitored by TLC performed in n-hexane/ethylacetate/dichloromethane (2/1/1, *v*/*v*/*v*) and visualized under UV at 254 nm. After the reaction, the mixture was adjusted to pH 7–8 with 10% NaHCO_3_ aqueous solution. Then, the mixture was stabilized to room temperature before 200 mL ice water was poured into it to cause precipitation. The separate precipitate was filtered, washed with water, dried, and crystallized to obtain 4.92 g of white, needle-shaped crystals with a yield of 98.4% [18].

### 3.3. Synthesis of Adamantane-1-Carbohydrazide (***3***)

A mixture of 4 g (20 mmoL) compound **2** and 25 mL (412 mmoL) 80% hydrazine hydrate aqueous solution were refluxed. The reaction was monitored by TLC using n-hexane/ethylacetate/dichloromethane (2/1/1, *v*/*v*/*v*) as the mobile phase and visualized under UV at 254 nm. At completion, 200 mL of ice water was added to the mixture. The separate precipitate was filtered and washed with water, dried to obtain 3.82 g of opalescent scaly solid with a yield of 95.5% [18].

### 3.4. Synthesis of Hydrazide-Hydrazones ***4a***–***i*** and ***5a***–***k***

A mixture of adamantane-1-carbohydrazide (**3**) and 1 mmoL of the appropriate aldehydes or ketones and 15 mL EtOH was stirred in reflux conditions. The reaction was monitored by TLC using the mobile phase as n-hexane/ethylacetate/dichloromethane (2/1/1, *v*/*v*/*v*) and visualized under UV at 254 nm. At completion, the solvent was removed under vacuum to yield a viscous liquid. It was allowed to stand at 0–5 °C to crystallize. Then, the crystal was filtered, washed with cold ethanol, and dried to obtain hydrazide-hydrazones **4a**–**i** and **5a**–**k**.

*N*′-(1-(4-hydroxyphenyl)ethylidene)adamantane-1-carbohydrazide (**4a**): M.p.: 252.5–254.1 °C, Yield 30.6%. ^1^H-NMR (500 MHz, DMSO-*d*_6_, *δ* ppm): 9.74 (1H, s, NH-N); 9.44 (1H, s, OH); 7.64 (2H, dd, *J1* = 2.0 Hz, *J2* = 7.0 Hz; Ar-H); 6.78 (2H, m, Ar-H); 2.16 (3H, s, CH_3_); 2.00 (3H, s, Adamantane-H); 1.90 (6H, s, CH_2 Ad_); 1.70 (6H, s, Adamantane-H). ^13^C-NMR (500 MHz, DMSO-*d*_6_, δ ppm): 172.6 (1C, CO); 158.8 (1C, Ar-C); 156.5 (1C, C=N); 128.9 (1C, C1); 127.9 (2C, Ar-C); 115.0 (2C, Ar-C); 38.2 (3C, Adamantane-C); 36.0 (3C, Adamantane-C); 27.6 (3C, Adamantane-C); 14.0 (1C, CH_3_). ESI-MS (*m*/*z*): [M − 1]^−1^ = 311.0; [M + 1]^+1^ = 313.1.

*N*′-(1-(4-nitrophenyl)ethylidene)adamantane-1-carbohydrazide (**4b**): M.p.: 226.0–227.6 °C; Yield 60.5%; ^1^H-NMR (500 MHz, DMSO-*d*_6_, *δ* ppm): 9.72 (1H, s, NH-N); 8.26 (2H, dd, *J1* = 2.0 Hz, *J2* = 7.0 Hz, Ar-H); 8.05 (2H, dd, *J1* = 2.0 Hz, *J2* = 7.5 Hz, Ar-H); 2.32 (3H, s, CH_3_); 2.01 (3H, s, Adamantane-H); 1.95 (6H, s, Adamantane-H); 1.71 (6H, s, Adamantane-H); ESI-MS (*m*/*z*): [M − 1]^−1^ = 340,0; [M + 1]^+1^ = 342,0.

*N*′-(1-(4-ethoxyphenyl)ethylidene)adamantane-1-carbohydrazide (**4c**): M.p.: 159.5–160.6 °C; Yield 32.2%; ^1^H-NMR (500 MHz, DMSO-*d*_6_, *δ* ppm): 9.47 (1H, s, NH-N); 7.74 (2H, d, *J* = 9.0 Hz, Ar-H); 6.95 (2H, d, *J* = 9,0 Hz, Ar-H); 4.06 (2H, dd, *J1* = 7.0 Hz; *J2* = 14.0 Hz, OCH_2_); 2.19 (3H, s, CH_3_); 2.00 (3H, s, Adamantane-H); 1.91 (6H, s, Adamantane-H); 1.70 (6H, s, Adamantane-H); 1.33 (3H, t, *J1* = 7.0 Hz, *J2* = 14.0 Hz, CH_3_). ESI-MS (*m*/*z*): [M + 1]^+1^ = 341.0.

*N*′-(1-(3-nitro-4-methoxyphenyl)ethylidene)adamantane-1-carbohydrazide (**4d**): M.p.: 182–184.1 °C; Yield 33.0%; ^1^H-NMR (500 MHz, DMSO-*d*_6_, *δ* ppm): 9.61 (1H, s, NH-N); 8.25 (1H, d, *J* = 2,5 Hz, Ar-H); 8.08 (1H, dd, *J1* = 2.0 Hz, *J2* = 8.5 Hz, Ar-H); 7.42 (1H, d, *J* = 7.0 Hz, Ar-H); 2.25 (3H, s, CH_3_); 2.00 (3H, s, Adamantane-H); 1.92 (6H, s, Adamantane-H); 1.71 (6H, s, Adamantane-H); ESI-MS (*m*/*z*): [M + 1]^+1^ = 372.0; [M − 1]^−1^ = 370.0.

*N*′-(1-(3-nitro-4-chlorophenyl)ethylidene)adamantane-1-carbohydrazide (**4e**): M.p.: 188.2–189.3 °C; Yield 26.2%. ^1^H-NMR (500 MHz, DMSO-*d*_6_, *δ* ppm): 9.72 (1H, s, NH-N); 8.39 (1H, d, *J* = 2.0 Hz, Ar-H); 8.09 (1H, dd, *J1* = 2.0 Hz, *J2* = 8.5 Hz, Ar-H); 7.83 (1H, d, *J* = 8.5 Hz, Ar-H); 2.29 (3H, s, CH_3_); 2.00 (3H, s, Adamantane-H); 1.94 (6H, s, Adamantane-H); 1.71 (6H, s, Adamantane-H). ESI-MS (*m*/*z*): [M + 1]^+1^ = 375.9; [M − 1]^−1^ = 373.9.

*N*′-(1-(3-nitro-4-chlorophenyl)ethylidene)adamantane-1-carbohydrazide (**4f**): M.p.: 190.7–191 °C. Yield: 29.0%; ^1^H-NMR (500 MHz, DMSO-*d*_6_, *δ* ppm): 9.59 (1H, s, NH-N); 7.76 (2H, d, *J* = 8.5 Hz, Ar-H); 7.63 (2H, d, *J* = 8.5 Hz, Ar-H); 2.25 (3H, s, CH_3_); 2.02 (3H, s, Adamantane-H); 1.94 (6H, s, Adamantane-H); 1.72 (6H, s, Adamantane-H); ESI-MS (*m*/*z*): [M + 1]^+1^ = 374.9.

*N*′-(1-(4-methoxyphenyl)ethylidene)adamantane-1-carbohydrazide (**4g**): M.p.: 171.6–173 °C; Yield 30.0%; ^1^H-NMR (500 MHz, DMSO-*d*_6_, *δ* ppm): 9.48 (1H, s, NH-N); 7.75 (2H, d, *J* = 8.5 Hz, Ar-H); 6.97 (2H, d, *J* = 9.0 Hz, Ar-H); 3.79 (3H, s, OCH_3_); 2.20 (3H, s, CH_3_); 2.00 (3H, s, Adamantane-H); 1.91 (6H, s, Adamantane-H); 1,70 (6H, s, Adamantane-H). ESI-MS (*m*/*z*): [M + 1]^+1^ = 327.0.

*N*′-(1-(4-methylphenyl)ethylidene)adamantane-1-carbohydrazide (**4h**): M.p.: 179.5–180.4 °C; Yield 37.3%; ^1^H-NMR (500 MHz, DMSO-*d*_6_, *δ* ppm): 9.48 (1H, s, NH-N); 7.75 (2H, d, *J* = 8.5 Hz, Ar-H); 6.97 (2H, d, *J* = 9.0 Hz, Ar-H); 3.79 (3H, s, OCH_3_); 2.20 (3H, s, CH_3_); 2.00 (3H, s, Adamantane-H); 1.91 (6H, s, Adamantane-H); 1.70 (6H, s, Adamantane-H); ESI-MS (*m*/*z*): [M + 1]^+1^ = 311.0.

*N*′-(1-phenyl)ethylidene)adamantane-1-carbohydrazide (**4i**): M.p.: 174.4–175.2 °C; Yield 54.5%; ^1^H-NMR (500 MHz, DMSO-*d*_6_, *δ* ppm): 9.54 (1H, s, NH-N); 7.79 (2H, m, Ar-H); 7.43–7.41 (3H, m, Ar-H); 2.24 (3H, s, CH_3_); 2.01 (3H, s, Adamantane-H); 1.93 (6H, s, Adamantane-H); 1.71 (6H, s, Adamantane-H). ESI-MS (*m*/*z*): [M + 1]^+1^ = 297.0.

*N*′-(4-hydroxybenzylidene)adamantane-1-carbohydrazide (**5a**): M.p.: 289.6–290.5 °C; Yield 44.0%; ^1^H-NMR (500 MHz, DMSO-*d*_6_, *δ* ppm): 10.56 (1H, s, NH-C); 9.86 (1H, s, OH); 8.28 (1H, s, N=CH); 7.49 (2H, d, *J* = 8.5 Hz, Ar-H); 6.82 (2H, d, *J* = 8.5 Hz, Ar-H); 2.01 (3H, s, Adamantane-H); 1.87 (6H, s, Adamantane-H); 1.71 (6H, s, Adamantane-H); ESI-MS (*m*/*z*): [M + 1]^+1^ = 298.9; [M − 1]^−1^ = 296.9.

*N*′-(4-ethoxybenzylidene)adamantane-1-carbohydrazide (**5c**): M.p.: 235.2–236.4 °C; Yield 15.1%; ^1^H-NMR (500 MHz, DMSO-*d*_6_, *δ* ppm): 10.61 (1H, s, NH-C); 8.32 (1H, s, N=CH); 7.57 (2H, d, *J* = 8.5 Hz, H2, H6); 6.97 (2H, d, *J* = 9.0 Hz, H3, H5); 4.07 (2H, m, OCH_2_); 2.00 (3H, s, CH
_Ad_); 1.86 (6H, s, CH_2 Ad_); 1.70 (6H, s, CH_2 Ad_); 1.34 (3H, t, *J1* = 7.0 Hz, *J2* = 14.0 Hz, CH_3_); ESI-MS (*m*/*z*): [M + 1]^+1^ = 327.0.

*N*′-(3-nitro-4-chlorobenzylidene)adamantane-1-carbohydrazide (**5e**): M.p.: 247.8–248.5 °C; Yield: 50.6%; ^1^H-NMR (500 MHz, DMSO-*d*_6_, *δ* ppm): 11.12 (1H, s, NH-C); 8.43 (1H, s, N=CH); 8.29 (1H, d, J = 2.0 Hz, Ar-H); 7.96 (1H, dd, *J1* = 2.0 Hz, *J2* = 8.5 Hz, Ar-H); 7.82 (1H, d, J = 8.5 Hz, Ar-H); 2.01 (3H, s, Adamantane-H); 1.87 (6H, s, Adamantane-H); 1.70 (6H, s, Adamantane-H); ESI-MS (*m*/*z*): [M + 1]^+1^ = 361.9; [M − 1]^−1^ = 359.9.

*N*′-(benzylidene)adamantane-1-carbohydrazide (**5i**): M.p.: 186.9–187.2 °C; Yield: 60.5%; ^1^H-NMR (500 MHz, DMSO-*d*_6_, *δ* ppm): 10.77 (1H, s, NH-C); 8.40 (1H, s, N=CH); 7.66–7.61 (2H, m, Ar-H); 7.45–7.40 (3H, m, Ar-H); 2.01 (3H, s, Adamantane-H); 1.88 (6H, s, Adamantane-H); 1.70 (6H, s, Adamantane-H); ESI-MS (*m*/*z*): [M + 1]^+1^ = 283.0.

*N*′-(2-hydroxy-5-methylbenzylidene)adamantane-1-carbohydrazide (**5j**): M.p.: 247.6–248.8 °C; Yield 60.4%; ^1^H-NMR (500 MHz, DMSO-*d*_6_, *δ* ppm): 11.15 (1H, s, OH); 11.08 (1H, s, NH-C); 8.48 (1H, s, N=CH); 7.23 (1H, d, J = 1.5 Hz, Ar-H); 7.07 (1H, dd, J1 = 2.0 Hz, J2 = 8.0 Hz, Ar-H); 6.79 (1H, d, J = 8.5 Hz, Ar-H); 2.23 (3H, CH_3_); 2.01 (3H, s, Ar-H); 1.87 (6H, s, Adamantane-H); 1.70 (6H, s, Adamantane-H); ESI-MS (*m*/*z*): [M + 1]^+1^ = 313.0; [M − 1]^−1^ = 310.9.

*N*′-(2,4-diemthylbenzylidene)adamantane-1-carbohydrazide (**5k**): M.p.: 283.5–284.0 °C; Yield 35.5%; ^1^H-NMR (500 MHz, DMSO-*d*_6_, *δ* ppm): 10.72 (1H, s, NH-C); 8.63 (1H, s, N=CH); 7.59 (1H, s, H6); 7.11 (2H, s, Ar-H); 2.36 (3H, s, CH_3_); 2.29 (3H, s, CH_3_); 2.02 (3H, s, Adamantane-H); 1.88 (6H, s, Adamantane-H); 1.71 (6H, s, Adamantane-H); ESI-MS (*m*/*z*): [M + 1]^+1^ = 311.0.

Spectra of synthesized compounds were displayed in Appendix A.

### 3.5. Dertemination of Antimicrobial Activity by the Dilution Method

The tested compounds **4a**–**i**, **5a**–**k** were dissolved in dimethylsulphoxide at the concentration of 100 mM, separately, to prepare the parent solutions. A total of 0.4 mL of the parent solution was withdrawn and 9.6 mL LB was added and mixed homogenously to obtain a working solution. Then, we prepared twofold serial dilution solutions in LB media at the concentrations of 100 μM, 50 μM, 25 μM, 12.5 μM, 6.25 μM, 3.13 μM, 1.56 μM, and 0.78 μM in 96 well plates, in triplicate. After that, 50 μL of microorganism suspension at 2 × 10^5^ CFU/mL was inoculated to each well and incubated at 37 °C for 24 h. Streptomycin and cycloheximide were used as the positive samples. Minimum inhibitory concentration (MIC) was determined as the lowest concentration that completely inhibited the growth of the microorganism, which was detected by the naked eye. Half of the maximal inhibitory concentration (IC_50_) is defined as the concentration of tested compounds that inhibits 50% visual growth of the test microorganism, which was determined by turbidity measurement on a Bitotech microplate reader and using RawData software [43].

### 3.6. Determination of Cytotoxicity Activity

The MTT (3-(4,5-dimethythiazol-2-yl)-2,5-diphenyl tetrazolium bromide) method was conducted to determine the cytotoxicity of the synthesized compounds **4a**–**i**, **5a**–**k** on human cancer cell lines, including hepatic cancer cell line Hep3B, human cervical cancer cell line HeLa, human lung cancer cell line A549, and human breast carcinoma MCF-7, according to a previous publication [43]. The cells, which were seeded at 5 × 10^6^ cell/well, were grown in 96-well plates with RPMI 1640 or DMEM media containing 10% fetal bovine serum (FBS), penicillin (100 IU/mL), and streptomycin (100 μg/mL) at 37 °C in a humid air incubator supplied with 5% CO_2_. The cells were allowed to grow for 24 h and then were removed to old media and treated with the sample. A total of 200 μL of the test samples was added to the well and incubated for 72 h in culture conditions. After removing the medium, 50 μL MTT solution (1 mg/mL in phosphate buffer saline) was then added to each well and the cells were further incubated at 37 °C for 4 h. After removing the MTT solution, 100 μL of isopropanol was added to each well. The absorbance was measured on a microplate reader (iMark™ Microplate Absorbance Reader, Bio-Rad Laboratories, CA, USA) at 570 nm. Suitable blank and positive controls (Camptothecine) were included. Cytotoxicity of the synthesized compounds was determined as the percent of cell survival, as follows: [OD (72 h)–OD (0 h)]/[OD(DMSO)–OD (0 h)]; where OD (72 h), OD (0 h), OD(DMSO) are the absorbance of test sample at 72 h, test sample at 0 h, and DMSO sample [42].

## 4. Conclusions

The synthesis and characterization of hydrazide-hydrazones with a 1-adamantane carbonyl moiety, **4a**–**i** and **5a**–**k**, were achieved. The synthesis was conducted by condensing adamantane-1-carbohydrazide (**3**) with substituted acetophenones to get compounds **4a**–**i**, and with substituted benzaldehydes to get compounds **5a**–**k**. The antimicrobial and cytotoxicity of the synthesized compounds were determined. The screening results revealed that compounds **4a**, **4d**, **5a**, and **5c** showed good antibacterial activity against Gram-positive bacteria and *Candida albicans* compared to known antibacterial and antifungal drugs. These compounds are considered as good candidates for new antimicrobial agents. Moreover, among the synthesized compounds, **4e** and **5e** displayed promising inhibitory activity against the growth of tested human cancer cell lines. Although the active compounds are considered to be good candidates as antimicrobial and antitumor agents, further studies, such as mechanism determination, should be undertaken.

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
