# Peer review of "Synthesis and Bioactivity of Hydrazide-Hydrazones with the 1-Adamantyl-Carbonyl Moiety"

_molecules, 2019, doi:10.3390/molecules24214000_

Round 1

Reviewer 1 Report

The paper by H.P. Van and coworkers entitled “Synthesis and Bioactivity of Hydrazide-hydrazone of 1-Adamantyl-carbonyl moiety” reports the antimicrobial and antifungal activities of some new adamantane derivatives. Moreover, they evaluated the cytotoxicity of the series towards 4 cancer cell lines.

By my opinion, the manuscript is interesting in particular for what concern the antimicrobial profile of the synthesized compounds. The supporting data is well written.

However, the paper lack of some fundamental concepts and need to be strongly improved before acceptance in Molecules journal.

Majors:

1- the authors did not explain any possible mechanism of action or even any potential molecular target of their compounds, both for antimicrobial activity and the cytotoxic activity. In this regard, molecular modeling support could be useful. The authors should post a discussion.

2- The cytotoxicity of compounds is very low, except for 4e and 5e, but only at 100 micromolar concentration so the statement that “..compounds 4e and 5e showed good inhibitory activity…” is a bit overrated and should be revised. In fact, since most of the compounds showed good viability towards all the cell lines, I would suggest to the authors to focus the rationale on antimicrobial properties and to use the viability data to enhance this point. For example, they can calculate the CC50 and the selectivity index-S.I. (calculated as CC50/MIC), and to reported these values in table 4 to highlight this point.

3- Since several hydrazide-hydrazones possess antimycobacterial activity (ref. 4, 8-10), an evaluation of the series of the synthesized compounds towards M. tuberculosis, would be of great interest.

4- the conclusions are too short and should provide more emphasis on the results and aim of the research.

Minors:

1- several typos to correct (i.e. lines 65 and 74: hydrazole instead of hydrazones; line 66 tablet instead of table; line 80 varrious instead of various; line 36-37 new molecular instead of new molecules; etc…).

2- table 1: since the crystallization solvent is the same for all the compounds, the 4th column could be deleted (and report in caption: *EtOH…)

3- in tables 2 and 4 values for the standards STM, CHM, CP should be reported in micromolar instead of micrograms/ml, for uniformity.

Reviewer 2 Report

This article reported the reaction of 1- adamantyl carbohydrazide with various substituted benzaldehydes and acetophenones to obtain hydrazide – hydrazones of 1-adamantyl carbonyl moiety and evaluate their biological activities. The screening result exhibited that some of these compounds with 1-adamantyl carbonyl moiety were founded as potential candidates against bacteria or possessed cytotoxicity activity against tested human cancer cell lines. Overall, the contribution is clearly written and all of works are almost well done. The manuscript is suitable for publication in this journal with following revisions. 

The authors should attempt to describe the correlations between functional groups and the activities against bacteria by chemical theories. Such as why the structure-cytotoxicity activity revealed that compounds 4e and 5e containing both of 3-NO2 and 4-Cl groups in benzyl ring have good cytotoxicity activity against four tested cancer cell lines ? Why compounds 4a, 4d, 5a and 5c were found as potential candidates against Gram-positive bacteria and Candida albicans?

The significant figures for weight and mole of starting material and product in experimental section should be consistent.

(i) A mixture of 5 g (2.7 mmoL) adamantane -1- carboxylic acid…obtain 4,92 g of white, needle-shaped crystals in yield of 98.4%.

(ii) A mixture of 4 g (0.02 moL) compound 2 and 25 ml (0.412 moL)…dried to obtain 3.82 g of opalescent scaly solid in yield of 95.5%.

Some spellings or typing are wrong.

(i) The spelling of “releved” is not correct in line 278 (The screening result releved that…)

(ii) “agaisnt Gram-possitve bacteria” is not correct in line 279

Reviewer 3 Report

The manuscript is well written and interesting and it deserves to be published, however I think that the biological activity of analyzed compounds should be also checked and analyzed in normal healthy cell line, at least for example in fibroblasts.

Besides that results are interesting and presented in a logical and coherent manner. Figures are readable and well described. Statistical analysis is clear.  

Round 2

Reviewer 1 Report

The paper by Van H.P. and coworkers has now been improved and can be publish in Molecules journal.